# Hallucination Detection for Generative Large Language Models by Bayesian Sequential Estimation

**Xiaohua Wang**[1,2], **Yuliang Yan**[1,2], **Longtao Huang**[3], **Xiaoqing Zheng**[1,2,*], **Xuanjing Huang**[1,2]

[1]School of Computer Science, Fudan University, Shanghai, China
[2]Shanghai Key Laboratory of Intelligent Information Processing
[3]Alibaba Group
{xiaohuawang22,ylyan21}@m.fudan.edu.cn
{zhengxq,xjhuang}@fudan.edu.cn

## Abstract

Large Language Models (LLMs) have made remarkable advancements in the field of natural language generation. However, the propensity of LLMs to generate inaccurate or non-factual content, termed "hallucinations", remains a significant challenge. Current hallucination detection methods often necessitate the retrieval of great numbers of relevant evidence, thereby increasing response times. We introduce a unique framework that leverages statistical decision theory and Bayesian sequential analysis to optimize the trade-off between costs and benefits during the hallucination detection process. This approach does not require a predetermined number of observations. Instead, the analysis proceeds in a sequential manner, enabling an expeditious decision towards "belief" or "disbelief" through a stop-or-continue strategy. Extensive experiments reveal that this novel framework surpasses existing methods in both efficiency and precision of hallucination detection. Furthermore, it requires fewer retrieval steps on average, thus decreasing response times[1].

## 1 Introduction

In the era of information overload and the proliferation of misleading or false information, automatic fact checking has become an essential tool for verifying the veracity of claims and combating misinformation. Large Language Models (LLMs), such as GPT-4 (OpenAI, 2023),PaLM (Chowdhery et al., 2022) and LLaMA (Touvron et al., 2023), have made significant advancements in the field of natural language generation (NLG). However, the inherent tendency of LLMs to generate inaccurate or non-factual content, commonly referred to as "hallucinations" (Bang et al., 2023; Ji et al., 2022), continues to present a significant challenge.

One previous work utilizes the sampled-based approach for detecting hallucination (Manakul

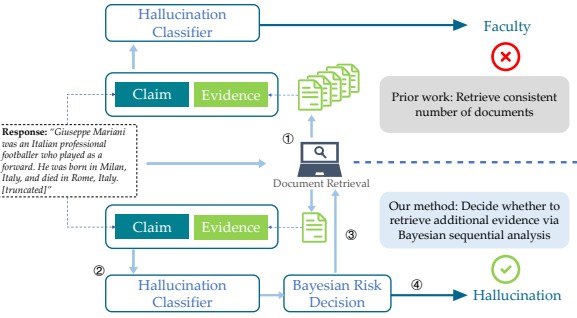

Figure 1: Evidence-based hallucination detection based on Bayesian sequential analysis. Prior works often retrieve a predetermined number of documents and overlooked the variability among different inputs. In our framework, (1) we first retrieve one document from the search engine; (2) External evidence and the claim generated by the LLM are input into the hallucination classifier to calculate the veracity score; Then we make a decision on whether to stop observing and make a veracity judgment based on the available evidence or to continue observing additional evidence. This decision is made based on the Bayesian risk decision and sequential analysis. (3) If stopping and making a determination carries a higher risk, we choose to continue observing additional evidence; (4) Otherwise, we choose to determine the veracity of the claim.

et al., 2023), they assume that LLMs are capable of producing similar responses and contain consistent facts related to a given concept. However, this method necessitates many samplings of the LLM, leading to tremendous costs in practical applications. Another commonly employed strategy in fact checking entails retrieving relevant evidence from various sources, such as texts (Thorne et al., 2018; Augenstein et al., 2019), knowledge graphs (Kim et al., 2023) and the webs (Chen et al., 2023; Kamoi et al., 2023) to validate or challenge claims. These methods usually retrieve a predetermined number of documents across diverse cases. However, the effectiveness of these approaches heavily depends on the selection of the number of documents to retrieve. While these methods can ensure

---

[1]Our code is available at https://github.com/xhwang22/HallucinationDetection.

a certain level of consistency, in practical applications, different claims may require varying amounts of external evidence for verification. This should be determined by the nature of the claim itself and the documents retrieved, rather than predetermined parameters. The optimal number of external evidence sources needed to validate a claim should be dynamically determined based on the specific context and characteristics of the claim, taking into account factors such as complexity, ambiguity, and the availability of relevant information. This study aims to investigate the feasibility of gradually collecting relevant information and adopting a data-driven approach to document retrieval in the context of hallucination detection for large language models (LLMs) in real-world scenarios. By employing such an approach, we can enhance retrieval efficiency and reduce unnecessary retrieval attempts.

In this paper, we propose a novel framework that leverages Bayesian sequential analysis (Wetherill, 1961; Arrow et al., 1949; Carlin et al., 1998) to enhance the retrieval efficiency for detecting whether claims generated by LLMs are hallucinated or factual in the wild. Recognizing that a single sentence may contain a mixture of factual and nonfactual information, we first decompose a claim into subclaims, each focusing on a single entity and its associated attributes (Kamoi et al., 2023). In the subsequent stage, we employ a search engine to retrieve relevant documents for each subclaim individually. These retrieved documents, in conjunction with their corresponding subclaims, are then input into a classifier to estimate their veracity (Nie et al., 2020). As illustrated in Figure 1, we consider the retrieved documents as a sequence and make informed decisions on whether additional information should be retrieved for a given case via Bayesian sequential analysis through a stop-or-continue strategy. Once the information gained is enough to decide whether claims are hallucinations, we stop and make an evaluation. Otherwise, we retrieve additional documents and make a decision in the next step. Finally, we aggregate the assessments of the subclaims to determine the overall veracity of the input claim.

To evaluate the effectiveness of our framework, we conducted experiments on the dataset of Self-CheckGPT (Manakul et al., 2023), containing $1,908$ sentences from $238$ articles generated by GPT-3, each labeled with its corresponding veracity. The experimental results demonstrate that our framework outperforms sample-based methods in terms of efficiency and accuracy in hallucination detection. In the task of sentence-level hallucination detection, our framework achieves comparable performance to the baseline approach. However, in the passage-level task, we observe an improvement of $6.43\%$ in terms of Spearman's correlation coefficient. Furthermore, across different sets of hyperparameter configurations, our framework consistently reduces the number of retrieved documents compared to the approach with a fixed number of retrieved documents. These results indicate the superiority of our approach in detecting hallucinations with improved efficiency and precision, showcasing its potential for practical applications.

## 2 Related Work

### 2.1 Hallucination Detection for Large Language Models

Several studies have explored the phenomenon of hallucination (Su et al., 2022; Lee et al., 2022; Dai et al., 2022) in large language models and have proposed various approaches for its causation and detection. Bang et al. (2023) conducted an evaluation of ChatGPT's hallucination on fact-checking test sets and knowledge-based QA tasks, highlighting the model's susceptibility to extrinsic hallucination compared to intrinsic hallucination (Ji et al., 2022).

In terms of detection, Kadavath et al. (2022) and Mündler et al. (2023) conducted a study to investigate whether language models can assess the validity of their own claims, aiming to explore the self-evaluative capabilities of language models. Azaria and Mitchell (2023) employed the internal state of LLMs to determine the truthfulness of statements by leveraging the hidden-layer activations as input to the classifier. Manakul et al. (2023) introduced a sample-based detection method for hallucination detection. They emphasized that when a language model is well-acquainted with a given concept, the sampled responses are likely to be similar and contain consistent facts. This method relies on analyzing the consistency of the generated samples to identify potential hallucinations. Unlike their approach, we employ an evidence-based fact-checking method, where we verify the correctness of claims based on external documents.

### 2.2 Evidence-based Fact Checking

Evidence-based fact-checking has gained significant attention as an effective approach to combat

misinformation and ensure information accuracy. FEVER (Thorne et al., 2018) utilized Wikipedia evidence for fact verification, while Augenstein et al. (2019) collected evidence from fact-checking websites and proposed a method that combines veracity predictions and evidence ranking.

Recent approaches include the use of knowledge graphs (KG) by Kim et al. (2023), where a classifier predicts relations and hops related to the claim, and related sequences are retrieved from the KG as evidence. Kamoi et al. (2023) put forward a dataset constructed using real-world claims and evidence. Their work focused on enhancing the performance of entailment models by reducing the complexity of claims through a decomposition process. By breaking down the claims into simpler components, they aim to facilitate a more effective evaluation of entailment and improve the overall performance of the models. Chen et al. (2023) presented an automated pipeline for fact-checking real-world claims, retrieving raw evidence from the web. However, these methods retrieve a fixed number of documents for all instances, whereas our work focuses on adaptively retrieving a varying number of documents for specific instances.

## 2.3 Bayesian Sequential Analysis

Sequential analysis (Wald, 1947) grounded in decision theory and Bayesian inference (Gunst and Shcherbakova, 2008; Box and Tiao, 1973), provides a valuable framework for making informed decisions based on accumulating evidence. The goal is to select decisions that minimize the expected loss, considering the associated costs. The general approach to Bayesian sequential decision problems involves backward induction (DeGroot, 2005; Berger, 2013), also known as stochastic dynamic programming (Ross, 2014). However, as the complexity of the problem increases, this method can become computationally intensive. To address this issue, Cardillo and Fu (1968) proposed an approximation by assuming that the subsequent stage is terminal, leading to a suboptimal solution. Expanding on the backward induction method, Brockwell and Kadane (2003) introduced a grid approximation approach. This technique estimates the expected loss at each decision time, significantly reducing computation time to a linear complexity relative to the number of decision stages.

In this paper, we utilize Bayesian sequential analysis to implement a stop-or-continue strategy for retrieving external documents. We treat the retrieved documents as a sequence and dynamically determine whether to continue or stop the retrieval process based on the accumulated evidence. This stop-or-continue strategy allows for efficient and effective information retrieval, optimizing the process by adapting to the evolving evidence and minimizing unnecessary retrieval attempts.

## 3 Method

In this paper, we propose a framework for detecting hallucinations of LLMs by leveraging external evidence.

Given a claim generated by the LLMs, we first decompose it into subclaims that contain basic knowledge and proceed to evaluate the veracity of each subclaim individually.

Subsequently, we employ a search engine to retrieve web documents for each subclaim as external evidence. Rather than predefining the number of documents to retrieve, we adopt a dynamic retrieval process that involves retrieving one document at one step and determining the need for additional retrieval based on the content of the retrieved document.

Then, we utilize a specific classification model to compute the entailment score between each document and subclaim which represents the extent to how well the evidence supports a given claim. The entailment score is processed into features, and a Naive Bayes classifier (NBC) is applied to calculate the probability of the subclaim's veracity based on these features.

Next, we adopt a stop-or-continue strategy, employing Bayesian risk decision to choose among three options: (1) stopping the retrieval process and classifying the subclaim as faculty, (2) stopping the retrieval process and classifying the subclaim as a hallucination, (3) or continuing to retrieve additional documents and make a choice in the next step. After making the decision to stop the retrieval process and determine the veracity or reaching the maximum retrieval times, we obtain the probability that the subclaim is faculty.

Finally, by aggregating the probabilities of the subclaims' veracity, we evaluate the overall veracity of the original claim and assess its potential as a hallucination.

In the following subsections, we discuss in more detail our method. In Section 3.1, we demonstrate the process of claim decomposition. In Section

3.2, we describe details of using a search engine to retrieve web documents. Section 3.3 explains how the entailment score is calculated and processed into features required by the Naive Bayes Classifier (NBC). In Section 3.4, we provide a specific description of the stop-or-continue strategy's details. Finally, in Section 3.5, we elaborate on how the veracity probabilities of subclaims are aggregated.

### 3.1 Claim Decomposition

Given a claim generated by LLMs, the direct veracity validation of the entire claim becomes challenging due to the presence of multiple knowledge instances within the generated content. So we first decompose the original claim into subclaims, with each subclaim specifically focusing on a single entity and its associated attributes. Kamoi et al. (2023) show that similar method allows for a more fine-grained analysis, enabling us to examine and evaluate each entity and its attributes individually. Simultaneously, we replace pronouns in sentences with their corresponding entities. This approach is implemented to enhance the retrieval of relevant evidence from web documents, making the search process more effective.

We employ a zero-shot prompt-based approach using the GPT-3.5 (`text-davinci-003`) model to perform claim decomposition and the full prompt can be seen in Appendix A. For sentences that have not been successfully decomposed, we solely perform entity replacements for pronouns.

### 3.2 External Documents Retrieval

To validate the real-world efficacy of our framework, we utilize the subclaims derived from the previous decomposition step to retrieve external documents from search engines. We input the subclaims as queries into the `Bing Search API`[2] to obtain URLs. We then utilize the `newspaper3k`[3] to extract web page content from the retrieved URLs. Note that the external documents are treated as a finite sequence with maximum length $K$ and are retrieved one by one. Therefore, we don't predetermine a fixed number of retrieval times. We disregard web content that is inaccessible, such as PDF files or protected websites.

---

[2]http://www.microsoft.com/en-us/bing/apis/
bing-web-search-api
[3]https://github.com/codelucas/newspaper

---

**Algorithm 1:** Hallucination detection

**Input:** $C$ : A claim generated by LLMs for hallucination detection;
    $\pi_1(0)$ : Initial probability of $C$ being factual;
    $K$ : Maximum retrieval times;
    $C_{FA}$ : Cost of false alarm;
    $C_M$ : Cost of miss;
    $C_{retrieve}$: Cost of retrieving an document;
**Output:** $P_{factual}(C)$: Probability of $C$ being factual;

1   $\{C^1, C^2, \cdots, C^L\} \leftarrow \text{ClaimDecompose}(C)$;
2   **for** $i \leftarrow 1$ *to* $L$ **do**
3     $n \leftarrow 1$;
4     **while** $n \leq k$ **do**
5       $E^n \leftarrow \text{RetrieveDocument}(C^i)$;
6       $f_n \leftarrow \text{CalEntailmentFeature}(E^n, C^i)$;
7       $\pi_1(n) \leftarrow \text{NBC}(\pi_1(n-1), f_n)$;
8       $R_{stop}(n) \leftarrow \min((1-\pi_1(n))C_M, (1-\pi_0(n))C_{FA})$;
9       $R_{continue}(n) \leftarrow C_{retrieve} + \mathbb{E}_{f_{n+1}}(R_{stop}(n+1))$;
10      **if** $R_{stop}(n) < R_{continue}(n)$ **then**
11        break;
12      **else**
13        $n \leftarrow n+1$;
14      **end**
15     **end**
16     $P^i_{factual} = \pi_1(n)$;
17 **end**
18 $P_{factual}(C) = \min_i P_{factual}(C^i)$;
19 **Return:** $P_{factual}(C)$

---

### 3.3 Entailment Score Calculation and Discretization

We utilize a specific classification model to calculate the entailment score between each document and subclaim which represents the extent to how well the evidence supports the given subclaim. A higher entailment score indicates stronger support from the external document for the corresponding subclaim. In this study, we use the DeBERTa-v3 (He et al., 2021) model fine-tuned on Natural Language Inference (NLI) dataset (Laurer et al., 2022) to perform a two-label classification (entailment and not entailment). Let $C$ refer to a subclaim and $E$ refer to a retrieved document, the entailment score $s(C, E)$ is the probability of the entailment label obtained by inputting the claim and evidence into DeBERTa-v3. Therefore, $s(C, E) \in (0, 1)$ and $s(C, E) \to 1$ if the evidence completely support the claim.

To overcome the limitation of the model's input length, we divide the document $E$ into text spans $\{E_1, E_2, \cdots, E_l\}$. Each text span comprises $m$ tokens, with a step size of $n$ ($n < m$) tokens. This means that we divide the document into text spans of length $m$ tokens and move the segmentation

window by $n$ tokens for each step. We calculate the entailment score between each text span and the corresponding subclaim, and select the highest entailment score as the entailment score for the original document:

$$s(C, E) = \max_i(s(C, E_i)) \qquad (1)$$

This approach ensures that the document's overall entailment score reflects the strongest evidence found within it.

**Score Discretization**

The entailment score is indeed a continuous value ranging between 0 and 1. To simplify the representation of the document characteristics, we transform the entailment score into a discrete entailment feature based on the assumption that two documents with similar entailment scores possess the same feature:

$$f(C, E) = \lfloor 10 \cdot s(C, E) \rfloor \qquad (2)$$

where $s$ is the continuous value of entailment score and $f \in \{0, 1, 2, \cdots, 8, 9\}$ is the discrete entailment feature given claim $C$ and evidence $E$. The entailment feature is then input into a Naive Bayes Classifier (NBC) to calculate the veracity of the claim in Equation 4.

### 3.4 Stop-or-Continue Retrieval Strategy

Given a subcliam $C$ that needs to be evaluated for veracity, and a finite sequence of retrieved external documents $\{E^1, E^2, \cdots, E^K\}$, we use the entailment features of each document to access the veracity $\theta$ of $C$. We denote $\theta = \theta_0$ to represent that $C$ is a hallucination and $\theta = \theta_1$ if $C$ is factual. At time $n < K$, we use $\pi_1(n)$ to represent the probability of $C$ being factual, given features $f_1, f_2, \cdots, f_n$:

$$\pi_1(n) = P(\theta = \theta_1 | f_{1:n}) \qquad (3)$$

where the $f_{1:n}$ is the entailment features of $E^{1:n}$ and $C$ calculated using Equation 2.

At step $n+1$, we use $\pi_1(n)$ as the prior probability and calculate the posterior probability $\pi_1(n+1)$ based on $\pi_1(n)$ and $f_{n+1}$:

$$\pi_1(n+1) = \frac{\pi_1(n)P(f_{n+1}|\theta_1)}{(1 - \pi_1(n))P(f_{n+1}|\theta_0) + \pi_1(n)P(f_{n+1}|\theta_1)} \qquad (4)$$

which comes by recursively applying Bayes' rule (Appendix B). This is the probability that $C$ is factual, given features $f_1, f_2, \cdots, f_n, f_{n+1}$, with an assumption of independence between the features. In our approach, we assume that the content generated by the large language model (LLM) has an equal prior probability of being hallucinations or factual information and set $\pi_1(0) = 0.5$.

One of the difficulties in calculating the above probability is the conditional probability $P(f_{n+1}|\theta_1)$ and $P(f_{n+1}|\theta_0)$. We use a sampling method to compute this term required in Equation 4. We sampled a set that consists of $s$ factual claims and $s$ nonfactual claims. For each claim, we retrieved a piece of external evidence and computed the entailment features of the claim and evidence. Then we estimate $P(f_{n+1}|\theta_1)$ and $P(f_{n+1}|\theta_0)$ by counting the factual and nonfactual claims (plus a smoothing term) that have the entailment feature equal to $f_{n+1}$.

**Bayesian Sequential Analysis**

For a subcliam $C$, external document comes one by one in sequence. At step $n < K$, we retrieve documents $\{E^1, E^2, \cdots, E^n\}$ and calculate the probability that $C$ is factual using Equation 4. Then we have three options:

- **Decide $\theta = \theta_1$:** We stop observing additional evidence and determine the subclaim $C$ is factual.
- **Decide $\theta = \theta_0$:** We stop observing additional evidence and determine the subclaim $C$ is a hallucination.
- **Keep test:** The previous evidence is not enough to evaluate the veracity of the subclaim $C$ and decide to retrieve additional documents.

If we choose the first two options, we stop observing additional external evidence. However, if we choose to continue with "**keep test**", we retrieve an additional document and make the choice among the three options again at step $n + 1$. We assume that the document sequence is finite, and when the maximum step $K$ is reached, we no longer choose "**keep test**".

We make decisions based on the minimization of Bayesian risk. In this study, we consider three possible risk costs:

- $C_{FA}$: Cost of false alarm, we declare $\theta = \theta_0$ but $C$ is factual. In this condition, we incur a cost when mistakenly classifying factual information as hallucinations.

- $C_M$: Cost of miss, we declare $\theta = \theta_1$ but $C$ is a hallucination. In this condition, we incur a cost when mistakenly classifying hallucinations as factual information.

- $C_{retrieve}$: Cost of retrieve an external evidence. In this work, we assume that the cost of retrieving external evidence is equal for each retrieval.

We choose to stop only when the cost of stopping is less than the minimum expected cost of retrieving one more document. If we stop and declare $\theta$, the cost due to "miss" would be $\pi_1(n)C_M$ and the cost due to "false alarm" would be $\pi_0(n)C_{FA}$. Therefore, if we make the decision to stop at step $n$, the expected risk $R_{stop}(n)$ is:

$$R_{stop}(n) = \min((1 - \pi_1(n))C_M, (1 - \pi_0(n))C_{FA}) \quad (5)$$

where $R_{stop}(n)$ is the expected cost of choose the stop option. If the cost due to "miss" is smaller than the cost due to "false alarm", we declare $\theta = \theta_1$ and determine that $C$ is factual. Otherwise, we declare $\theta = \theta_0$ and determine that $C$ is a hallucination.

If we choose to retrieve an additional document and make a choice in the next step, because we do not know the retrieved document $E^{n+1}$ at step $n$, so we calculate the minimum expected risk:

$$R_{continue}(n) = C_{retrieve} + \sum_{f_{n+1}=0}^{9} R(n+1) \cdot P(f_{n+1}) \quad (6)$$

where $f_{n+1}$ is the entailment feature of $E^{n+1}$ and $C$. $R_{continue}$ is the minimum expected risk to retrieve an additional document and make a choice in the next step. $R$ is the overall minimum expected cost of the three options:

$$R(n) = \min(R_{continue}(n), R_{stop}(n)) \quad (7)$$

We use this recursive equation to obtain the optimal solution. We stop and determine the probability of $C$ being factual is $\pi_1(n)$ and evaluate the veracity of $C$ if this action results in minimum expected cost $R_{stop}(n)$. Specifically, if $R_{stop}(n)$ results in the cost of miss, we consider $C$ to be factual. Otherwise, we consider $C$ to be a hallucination. On the other hand, if the measurement of $R_{continue}(n)$ yields the minimum expected cost, the process continues and an additional document $E^{n+1}$ is retrieved.

To obtain the optimal Bayesian decision in the finite sequential setting at any step other than the last, one must consider the probability of data values that have not yet been observed. This requires working the problem backward (from the max step $K$ back to time 0), a process referred to as backward induction (DeGroot, 2005; Berger, 2013). However, computing Equation 6 is a bit awkward, as we need to consider all discrete values for subsequent steps. The computation complexity that implements the backward induction solution increases exponentially in the number of backward steps. Therefore, we use the truncation approximation made one-step-ahead (Cardillo and Fu, 1968) which leads to the suboptimal solution. In each step, the choice is made based on the assumption that the next step is the final step, meaning that only the first two options are considered. In this case, Equation 5 still hold, but $R(n + 1)$ is replaced by $R_{stop}(n+1)$ in Equation 6:

$$R_{continue}(n) = C_{retrieve} + \sum_{f_{n+1}=0}^{9} R_{stop}(n+1) \cdot P(f_{n+1})$$
$$(8)$$

By using this approach, the computational complexity decreases from exponential to linear. Specifically, it reduces from $O(10^K)$ to $O(K)$.

## 3.5 Aggregation

Finally, we calculate the original claim's probability of factual by considering the minimum probability of factual among the subclaims:

$$P_{factual}(C) = \min_i P_{factual}(C^i) \quad (9)$$

Where $P_{factual}$ is the probability of factual and the original claim $C$ is decomposed into $L$ subclaims $\{C^1, C^2, \cdots, C^L\}$. This is based on the assumption that if any portion of the original claim is nonfactual, we consider the whole sentence to be a hallucination.

## 4 Experiments

We conducted experiments on the hallucination detection classification task. First, we compared our framework with SelfCheckGPT to evaluate the effectiveness of our approach in hallucination detection at both the sentence-level and passage-level. Next, we performed a comparative experiment using a fixed number of retrieved documents as external evidence to validate the efficiency of our framework in improving retrieval efficiency and reducing the number of external evidence. Finally, we conducted ablation study to evaluate the necessity of the claim decomposition step.

| Method | | Sentence-level (AUC-PR) | | | | Passage-level (Corr.) | |
|---|---|---|---|---|---|---|---|
| | | Evidence Num | Nonfact | Factual | Acc | Pearson | Spearman |
| Self-Detection | | - | - | - | 31.01 | - | - |
| SelfCheckGPT | w/ BERTScore | 20 | 81.96 | 44.23 | - | 58.18 | 55.90 |
| | w/ QA | 20 | 84.26 | 48.14 | - | 61.07 | 59.29 |
| | w/Unigram (max) | 20 | 85.63 | 58.47 | - | 64.71 | 64.91 |
| | Combination | 60 | **87.33** | 61.83 | - | 69.05 | 67.77 |
| Our Framework | $C_M = 14, C_{FA} = 24$ | 3.05 | 82.42 | 57.01 | 80.24 | 71.37 | 64.55 |
| | $C_M = 28, C_{FA} = 96$ | 6.22 | 86.45 | **61.96** | **82.39** | **81.18** | **74.20** |

Table 1: Follow prior work (Manakul et al., 2023), we report the Area Under the Precision-Recall Curve (AUC-PR) at the sentence-level and the Pearson and Spearman's rank correlation coefficient w.r.t human judgments at the passage-level. Additionally, we report the number of external evidence. For SelfCheckGPT, "Evidence Num" represents the number of samples obtained from GPT-3. For our framework, "Evidence Num" represents the average number of retrieved external documents. Furthermore, for our framework, we also report the sentence-level accuracy. BERTScore, QA and Unigram are three metrics to evaluate the consistency of samples in SelfCheckGPT.

## 4.1 Experiment setup

**Dataset** We use the dataset of SelfCheckGPT (Manakul et al., 2023) which contains $1,908$ sentences from 298 article generated by GPT-3. Each sentence is labeled with one of the three veracity labels: **"Accurate"**, **"Minor Inaccurate"** and **"Major Inaccurate"**. In the experiments, "Major Inaccurate" and "Minor Inaccurate" sentences are labeled as non-factual class, while sentences labeled as "Accurate" are considered factual class.

**Baseline** SelfCheckGPT samples from GPT-3 multiple times and utilizing consistency between samples to identify hallucinations. We use SelfCheckGPT as the baseline to evaluate the performance of our framework for hallucination detection at sentence-level and passage-level. We also query ChatGPT itself to determine whether it constitutes hallucination as an additional baseline(Mündler et al., 2023).

**Hyperparameters Settings** For calculation of the conditional probability of Equation 4, we used $s = 200$ and a smoothing term of 1 in the experiments reported here. In section 4.3, we adjust $C_{FA}$ and $C_M$ on our framework for hyperparameter experiments. For the experiments conducted outside of that section, we set the value of the three possible costs of Bayesian sequential analysis $C_{FA} = 24$, $C_M = 14$ and $C_{retrieve} = 1$. For the length of text spans and the step size during document segmentation, we set $m = 400$ and $n = 100$. Additionally, we set the maximum retrieval time $K = 10$.

**Evaluation Metrics** Following SelfCheckGPT, we report AUC-PR for sentence-level detection tasks. For passage-level detection, we calculate the average score of all sentences in a passage to obtain the passage's veracity score. Passage-level ranking performances are measured by Pearson correlation coefficient and Spearman's rank correlation coefficient w.r.t. human judgments.

For our framework, we also report average search time and accuracy (ACC) for sentence-level detection tasks. To validate the advantage of our method in terms of sample cost, we report the number of external evidence. For SelfCheckGPT, we consider the number of times they perform sampling from GPT-3 as the quantity of external evidence. For our framework, we report the average number of retrieved documents at the sentence-level. To validate our advantage in terms of sampling cost, we assume that SelfCheckGPT incurs the same cost for one GPT-3 sample as it does for searching an external document.

## 4.2 Results of Hallucination Detection

The results of our experiments can be found in Table 1, which provides an overview of the performance of our framework in hallucination detection. When the average number of retrieved documents per sentence is 3.05, our framework outperforms SelfCheckGPT with single consistency metric, which rely on 20 external evidence, in terms of passage-level performance. As the average number of retrieved documents increased to 6.22, our framework shows slightly lower performance in Nonfact detection but uses less external evidence. In Factual detection and passage-level evaluation, we achieve significant improvements over the baseline. These results demonstrate the effectiveness of our framework in hallucination detection.

## 4.3 Effectiveness for Reducing Retrieval Times

In this section, we conducted comparative experiments using a fixed number of retrieved documents.

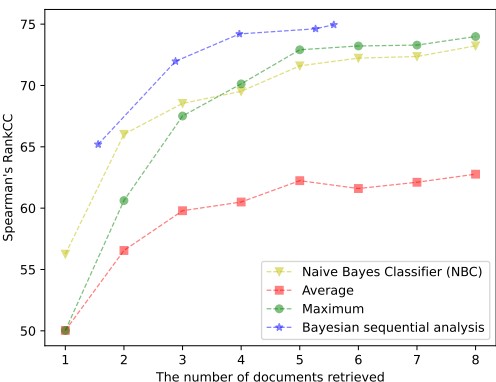

Figure 2: The performance of Bayesian sequential analysis method on ranking passages (Spearman's) versus strategies with fixed number of retrieval documents.

For the case of using a fixed number of retrieved documents, we considered three methods to aggregate the features of these documents:

- Naive Bayes Classifier (NBC): We treated the documents as a fixed number of features and used a Naive Bayes classifier, as described in Eq.4, for hallucination detection.

- Average: For each document, we compute an entailment score and then average the scores of all the documents to obtain the veracity score of the claim.

- Maximum: We calculated an entailment score for each document and selected the highest score as the veracity score for the sentence. This approach is based on the assumption that hallucinations usually lack any supporting evidence in external documents. Therefore, if there are documents that can validate a sentence, we consider the sentence factual.

The $C_M$ and $C_{FA}$ hyperparameters represent the costs involved in making Bayesian risk decisions. Different values of $C_M$ and $C_{FA}$ can lead to different decisions by the strategy, for the same $\pi_1(n)$ and $C_{retrieve}$, thereby influencing the average number of retrieved documents. To evaluate the effectiveness of our framework in improving retrieval efficiency and reducing search frequency, we conduct comparative experiments using the fixed number of retrieved documents. We randomly select five sets of $C_M$ and $C_{FA}$ values and compare the results with the fixed search times approach.

The results obtained from the experiments are presented in Figure 2, which illustrates the relationship between the average number of retrieved documents in our framework and the Spearman's correlation coefficient at the passage-level hallucination detection. Based on the results, we observe that as the average number of retrieved documents in our framework approaches the fixed number of retrievals, there is a higher Spearman's correlation coefficient at the passage-level hallucination detection. This implies that our framework achieves better performance in hallucination detection with fewer retrievals compared to the fixed number of retrievals. These findings indicate that our framework effectively reduces the number of retrievals required while maintaining or even improving the performance of hallucination detection. This demonstrates the efficacy of our approach in enhancing retrieval efficiency and reducing search frequency in the context of hallucination detection tasks.

| Method | Sentence-level (AUC-PR) | | |
|---|---|---|---|
| | Nonfact | Factual | Acc |
| w/o Decomposition | 80.04 | 53.71 | 79.19 |
| w Decomposition | **82.42** | **57.01** | **80.24** |

Table 2: The performance at the sentence-level by directly using the original sentence for hallucination detection.

## 4.4 The Effect of Claim Decomposition

In this section, we validate the contribution of the claim decomposition step to the overall performance. We consider the scenario where only the decomposed subclaims are used for document retrieval and directly perform hallucination detection on the original sentence.

The results in Table 2 demonstrate that when subclaims are only used for document retrieval, the performance of the framework decreases to some extent at both the sentence level and passage level. This indicates that directly determining the veracity of a sentence that contains multiple pieces of knowledge is challenging. It highlights the necessity of the problem decomposition step.

## 5 Conclusions

In this study, we propose a framework that utilizes Bayesian sequential analysis for detecting hallucinations in large language models (LLMs). We consider the retrieved documents as a sequence and employ a stop-or-continue strategy to make in-

formed decisions on whether to retrieve additional information for a given case. By applying Bayesian sequential analysis, our framework achieves competitive results on annotated GPT-3 responses, surpassing sample-based methods and retrieval strategies with a fixed number of search documents. Additionally, we validate the practical applicability of our framework by conducting searches for relevant documents on the web, demonstrating the feasibility of our approach in corroborating the claims made by LLMs with real-world information.

## Limitations

In this study, the implemented strategy amalgamates individual pieces of information extracted from a search engine at each iterative stage. An inherent limitation of this methodology is the potentially extended duration required for inference. For future studies, it would be better to consider the integration of information drawn from multiple documents concurrently, which could significantly enhance the speed of the inference process.

## Acknowledgements

The authors would like to thank the anonymous reviewers for their valuable comments. This work was supported by National Natural Science Foundation of China (No. 62076068), and Shanghai Municipal Science and Technology Project (No. 21511102800).

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

## A  Claim Decomposition Prompts

The zero-shot prompt we use to decompose the claim is shown in Figure 3.

Follow these steps to process the text in the triple delimiter:

Step 1: Rewrite the text while use the original names of entities instead of pronouns when referring to them.

Step 2: Decompose the text into basic knowledge, requiring each piece of knowledge to contain only one entity and one attribute of that entity and one value of the attribute.

Step 3: Express each decomposed piece of knowledge in a natural language sentence.

Step 4: Output which sentence in step three was obtained from the original text.

Output the result in JSON format, the key of JSON is 'step1', 'step2', 'step3' and 'step4', the value of step3 should be a list of sentences and the value of step4 should be pairs of sentence in step3 and original sentence in the original text and the index of the sentence in the original text.

```{ passage }```
Output:

Figure 3: The zero-shot prompt for decompose the sentence of the passage into subclaims.

## B  Derivation of Equation 4

Given a subcliam $C$ that needs to be evaluated for veracity, and a finite sequence of retrieved external documents $\{E^1, E^2, \cdots, E^K\}$, we use the entailment features of each document to access the veracity $\theta$ of $C$. We denote $\theta = \theta_0$ to represent that $C$ is a hallucination and $\theta = \theta_1$ if $C$ is factual. At time $n < K$, we use $\pi_1(n)$ to represent the probability of $C$ being factual, given features $f_1, f_2, \cdots, f_n$:

$$\pi_1(n) = P(\theta = \theta_1 | f_{1:n}) \qquad (10)$$

where the $f_{1:n}$ is the entailment features of $E^{1:n}$ and $C$ calculated using Equation 2.

$$\pi_1(n+1) \tag{11}$$

$$= P(\theta = \theta_1 | f_{1:n+1}) \tag{12}$$

$$= \frac{P(\theta_1, f_{1:n+1})}{P(f_{1:n+1})} \tag{13}$$

$$= \frac{P(\theta_1)P(f_{1:n+1}|\theta_1)}{P(f_{1:n+1}|\theta_1)P(\theta_1) + P(f_{1:n+1}|\theta_0)P(\theta_0)} \tag{14}$$

$$= \frac{P(\theta_1)\Pi_{i=1}^{n+1}P(f_i|\theta_1)}{P(\theta_1)\Pi_{i=1}^{n+1}P(f_i|\theta_1) + P(\theta_0)\Pi_{i=1}^{n+1}P(f_i|\theta_0)} \tag{15}$$

$$= \frac{P(f_{n+1}|\theta_1)P(f_{1:n}, \theta_1)}{P(f_{n+1}|\theta_1)P(f_{1:n}, \theta_1) + P(f_{n+1}|\theta_0)P(f_{1:n}, \theta_0)} \tag{16}$$

$$= \frac{P(f_{n+1}|\theta_1)P(\theta_1|f_{1:n})}{P(f_{n+1}|\theta_1)P(\theta_1|f_{1:n}) + P(f_{n+1}|\theta_0)P(\theta_0|f_{1:n})} \tag{17}$$

$$= \frac{\pi_1(n)P(f_{n+1}|\theta_1)}{(1 - \pi_1(n))P(f_{n+1}|\theta_0) + \pi_1(n)P(f_{n+1}|\theta_1)} \tag{18}$$

Deriving Equation 15 from Equation 14 based on the assumption of independence between the features.

## C   Self-Detection Prompts

For querying ChatGPT itself to see whether it's hallucination, we use the prompt shown in Figure 4.

I give you one statement, please conclude whether the statement is nonfactual with Yes or No.

Statement: {Hypothesis}

Figure 4: The zero-shot prompt for querying ChatGPT itself to see whether it's hallucination.