# OpenReview forum: "Hallucination Detection for Generative Large Language Models by Bayesian Sequential Estimation"
_EMNLP/2023/Conference — EMNLP 2023 Main_

### Official Review · Reviewer_o6Kk · 2023-08-01

**Soundness:** 3

**Excitement:**

3: Ambivalent: It has merits (e.g., it reports state-of-the-art results, the idea is nice), but there are key weaknesses (e.g., it describes incremental work), and it can significantly benefit from another round of revision. However, I won't object to accepting it if my co-reviewers champion it.

**Paper Topic And Main Contributions:**

The paper aims to tackle the hallucination detection problem. The authors first decompose a claim into subclaims using GPT-3.5. They then employ a search engine (Bing Search API and newspaper3k) to retrieve relevant documents for each subclaim individually. Subsequently, they calculate the subclaim's veracity using an NLI model (DeBERTa-v3). Next, they adopt a stop-or-continue strategy to decide whether to stop the evaluation of a subclaim or not. Finally, they evaluate the overall veracity of the original claim and assess its potential as a hallucination.

----Post-rebuttal-----

The author has only partially addressed my concerns; I am still concerned about the comparison of detection times. I can only improve my soundness score to 3.

**Reasons To Accept:**

- The paper presents good experimental results.
- The proposed Bayesian sequential analysis for hallucination detection is novel, and I haven't seen it before.

**Reasons To Reject:**

- The proposed method is complex and might not be easy to reproduce. For example, the method relies on GPT-3.5, Bing Search API, DeBERTa-v3, and Naive Bayes Classifier. The sensitivity of the method to some key hyper-parameters, such as $C_M$ and $C_{FA}$, is also not presented.
- The complexity of the method might hinder the efficiency of the detection process. While the authors indicate that their method requires fewer document retrievals, it would be valuable to directly compare the detection time with alternative approaches to assess its efficiency in practice.

**Reproducibility:**

3: Could reproduce the results with some difficulty. The settings of parameters are underspecified or subjectively determined; the training/evaluation data are not widely available.

**Reviewer Confidence:**

3: Pretty sure, but there's a chance I missed something. Although I have a good feel for this area in general, I did not carefully check the paper's details, e.g., the math, experimental design, or novelty.

---

> ### Author Rebuttal · Authors · 2023-08-27
>
> Thank you for your insightful and valuable comments.
>
> Q1: The proposed method is complex and might not be easy to reproduce. For example, the method relies on GPT-3.5, Bing Search API, DeBERTa-v3, and Naive Bayes Classifier. The sensitivity of the method to some key hyper-parameters, such as C_FA and C_M, is also not presented.
>
> R1: We are committed to releasing our source code and data to the public along with thorough and detailed documentation. We would like to emphasize that each of the components we present is indispensable in realizing effective hallucination detection within the framework of Bayesian sequential analysis, although there is flexibility in terms of their specific implementation.
>
> We propose the following procedure to determine the values for the hyperparameters C_FA and C_M. Initially, a subset of k samples (where k = 3 in our experiments) is randomly selected, and subsequently, manual annotations are applied to capture the claims, evidence, and decisions from the stop-or-continue trial. Through this process, we refine the potential ranges within which C_FA and C_M values lie.
>
> The following table show the means and variances of sentence-level AUC-PR and passage-level correlation. These metrics are evaluated across five distinct pairings of C_FA and C_M, selected randomly within their refined intervals (i.e., about several hundred pairs from C_FA = [10, 50] and C_M = [25, 100]). Our analysis reveals that the impact of the two hyperparameters on performance appears to be relatively insensitive when sampled randomly from the specified ranges, as determined by a limited number of initial samples. It is noteworthy that this study employed only three samples to establish the initial ranges for C_FA and C_M. However, it is important to note that in practical scenarios, increasing the number of samples for range determination leads to narrower intervals and consequently enhanced performance.
>
> |       | Evidence Num| Nonfact|Factual|Accuracy|Pearson|Spearman|
> |:-----------:| :-------------:|:-------------:|:-------------:|:-------------:|:-------------:|:-------------:|
> | $\text{Average}_{\text{var}}$|$\text{5.97}_{\text{0.06}}$|$\text{86.37}_{\text{0.13}}$|$\text{62.35}_{\text{0.07}}$|$\text{82.19}_{\text{0.31}}$|$\text{80.80}_{\text{0.11}}$|$\text{73.65}_{\text{0.57}}$|
>
> Q2: The complexity of the method might hinder the efficiency of the detection process. While the authors indicate that their method requires fewer document retrievals, it would be valuable to directly compare the detection time with alternative approaches to assess its efficiency in practice.
>
> R2: Incorporating the assessment of whether the outputs from a Language Model (LLM) are hallucinatory or not will undoubtedly introduce additional costs. These incremental costs primarily encompass both the frequency of LLM interactions (e.g., ChatGPT) and the number of webpages retrieved. The goal of this study is to optimize the trade-off between costs and benefits during the hallucination detection process within the framework of Bayesian sequential analysis.
>
> A closely related work, SelfCheckGPT (Manakul et al., 2023), necessitates an average of 60 interactions with ChatGPT for each input, as opposed to a solitary interaction required for the decomposition process in our approach. Furthermore, our method needs an average retrieval count of 6.22 webpages. The computational overhead associated with webpage retrieval is much lower compared to the expenditure involved in ChatGPT interactions. Therefore, our method reveals superior computational efficiency when contrasted with the baseline. In Section 4.3, we also show that our method achieves comparable performance with a reduced number of retrievals. This affirms the efficacy of our approach in achieving the desired outcomes in a resource-efficient manner.

---

### Official Review · Reviewer_rSTF · 2023-08-01

**Soundness:** 3

**Excitement:**

4: Strong: This paper deepens the understanding of some phenomenon or lowers the barriers to an existing research direction.

**Paper Topic And Main Contributions:**

This paper propose an LLM hallucination detection framework leveraging external evidence. The pipeline includes breaking LLM generations into sub-claims, dynamic evidence retrieval using search engine, a classification module to determine how likely the evidence supports the claims, coupled with a stop-or-continue strategy based on Bayesian risk decision. The authors conduct experiments on the SelfCheckGPT dataset and find that the detection accuracy can be increased a lot with fewer retrieved documents. Some ablations also demonstrate the effectiveness of Bayesian risk decision in thie pipeline.

**Reasons To Accept:**

- Good intuition to restrict the number of retrieved documents and use Bayesian risk decision.
- Solid writing and introduction to the whole methodologies.
- Experimental results seem promising.

**Reasons To Reject:**

- The entailment scores are still calculated using another deep learning model, which can still be silver-standard. It would be better to point out how reliable Deberta-v3-NLI is in terms of your task, with some expert annotations.
- 2.1 seems weak. There should be many 2023 works for hallucination detection. Also there're only limited baselines in Table 1. What about some intuitive baselines like querying ChatGPT itself to see whether it's hallucination? This should be a strongo one.

**Reproducibility:**

3: Could reproduce the results with some difficulty. The settings of parameters are underspecified or subjectively determined; the training/evaluation data are not widely available.

**Reviewer Confidence:**

3: Pretty sure, but there's a chance I missed something. Although I have a good feel for this area in general, I did not carefully check the paper's details, e.g., the math, experimental design, or novelty.

---

> ### Author Rebuttal · Authors · 2023-08-27
>
> Thank you for your insightful and valuable comments.
>
> Q1: The entailment scores are still calculated using another deep learning model, which can still be silver-standard. It would be better to point out how reliable Deberta-v3-NLI is in terms of your task, with some expert annotations.
>
> R1: This is a very good suggestion. In response, we conducted a comprehensive evaluation of Deberta-v3-NLI as per your recommendation. Our evaluation encompassed multiple established natural language inference (NLI) datasets that are widely recognized in the field. In addition to these benchmark datasets, we created an additional dataset comprising 300 examples, designated as "our_test" This new dataset was created by randomly selecting 150 claims from our test set, and for each of these selected claims, we thoughtfully gathered both supporting and opposing texts. Note that Deberta-v3-NLI was not fine-tuned on the datasets by our team throughout this evaluation process. The accuracy of Deberta-v3-NLI is listed in the table below.
> | Dataset| mnli_test_m|mnli_test_mm|anli_test|anli_test_r3|ling_test|wanli_test|our_test|
> |:-----------:| :-------------:|:-------------:|:-------------:|:-------------:|:-------------:|:-------------:|:-------------:|
> | Accuracy|0.912|0.908|0.702|0.64|0.87|0.77|0.823|
>
> It is crucial to highlight that we have taken into account the unreliability of Deberta-v3-NLI's predicted entailment scores within the framework of Bayesian sequential analysis. In our approach, these predicted scores serve as priors for the stop-or-continue retrieval strategy, which are then subject to gradual refinement through the accumulation of newly retrieved evidence, resulting in the establishment of more robust posterior probabilities. This integration of a Bayesian framework significantly boosts the performance on hallucination detection, effectively addressing the initial uncertainties associated with Deberta-v3-NLI's entailment predictions.
>
> Q2: 2.1 (baselines) seems weak. There should be many 2023 works for hallucination detection. Also there're only limited baselines in Table 1. What about some intuitive baselines like querying ChatGPT itself to see whether it's hallucination? This should be a strong one.
>
> R2: In the section of related work, we have discussed nearly all pertinent studies. Yes, there would be many 2023 works for hallucination detection. It is noteworthy that a considerable portion of these works became available either shortly after our submission or in the days immediately preceding it.
> Taking into account your valuable suggestion, we proceeded to execute an experiment based on the baseline you recommended. It employs ChatGPT itself to assess the presence of hallucination, akin to the method outlined in [1]. The experimental results are listed in the following table, and the results indicated by "Self-detection" highlight a notably inferior performance (drop about 0.5 in accuracy) by ChatGPT on hallucination detection. This finding aligns with a recent investigation [1], wherein the authors asserted that "The latest LMs still cannot handle factuality directly without relying on grounded knowledge."
>
> |     Method  | Sentence-level Accuracy|
> |:-----------:| :-------------:|
> | Self-Detection|31.01|
> | Our framework|82.39|
>
> Reference:
> [1] Niels Mündler, Jingxuan He, Slobodan Jenko, and Mar-tin T. Vechev. 2023. Self-contradictory hallucinations of large language models: Evaluation, detection and mitigation. ArXiv, abs/2305.15852.

---

### Official Review · Reviewer_6dtQ · 2023-08-04

**Soundness:** 4

**Excitement:**

4: Strong: This paper deepens the understanding of some phenomenon or lowers the barriers to an existing research direction.

**Paper Topic And Main Contributions:**

The paper proposes  a framework for detecting hallucinations of LLMs by leveraging external evidence. Specifically, given some text generated by the LM, a number of supporting documents is retrieved via web search. The number of retrieved documents is chosen by Bayesian inference, allowing to tailor it to the degree of confidence at this stage. Specifically, at each stage it possible to decide whether to continue, stop and classify the claim as false, or stop and classifying it as true. The decision is based on features derived from a naive-bayes classifier predicting the veracity given the retrieved document and the claim generated by the LM.

**Reasons To Accept:**

The paper applies an interesting technique to an important problem in the era of large LMs: claim verification and hallucination detection. The paper is clearly written, and provides a good introduction to the employed techniques for those without prior knowledge.

**Reasons To Reject:**

I do not see major issues with this paper. It can be interesting to apply the method on simpler claims, that do not require the decomposition step; although the paper does try to evaluate the effect of this step.

**Reproducibility:**

4: Could mostly reproduce the results, but there may be some variation because of sample variance or minor variations in their interpretation of the protocol or method.

**Reviewer Confidence:**

3: Pretty sure, but there's a chance I missed something. Although I have a good feel for this area in general, I did not carefully check the paper's details, e.g., the math, experimental design, or novelty.

---

> ### Author Rebuttal · Authors · 2023-08-27
>
> Thank you for your insightful and valuable comments.
>
> Q1: It can be interesting to apply the method on simpler claims, that do not require the decomposition step; although the paper does try to evaluate the effect of this step.
>
> R1: We have considered your feedback regarding the necessity of decomposition for certain simpler claims. In response to your suggestion, we conducted a thorough examination of the impact of decomposition on simple claims. The results indicate that, when dealing with randomly selected 200 simple claims, there was no evidence of excessive decomposition. During our investigation, approximately 60% of instances remained in an undecomposed state due to their inherent simplicity, aligning with the predefined decomposition criteria specified by the prompt " focusing on a single entity and its associated attributes "
>
> However, it is worth noting that we did identify some instances where appropriate decomposition should have occurred but did not manifest despite its expected applicability. Our module exhibits a bias towards avoiding over-decomposition.
>
> In Section 4.4 of our paper, we delve into an ablation study that demonstrates the critical role of the decomposition step within the proposed approach. This step significantly enhances performance by a significant margin of 3.3% in terms of accuracy.

---

### Meta-Review · Area_Chair_6CYE · 2023-09-26

**Recommendation:** 5

**Metareview:**

The paper presents a Bayesian-based method for hallucination detection in LLMs. Reviewers agree on the merit of the proposal and mostly raise minor issues.

---

### Decision · Program_Chairs · 2023-10-07

**Decision:**

Accept-Main

**Comment:**

The paper presents a Bayesian-based method for hallucination detection in LLMs. Reviewers agree on the merit of the proposal and mostly raise minor issues.